# Deeply Integrated GNSS/Gyro Attitude Determination System

**DOI:** 10.3390/s20082203

**Published:** 2020-04-13

**Authors:** Alexander Perov, Alexander Shatilov

**Affiliations:** Navigation Systems Laboratory, Moscow Power Engineering Institute, Kranokazarmennaya street 14, Moscow 111250, Russia

**Keywords:** attitude, navigation, GNSS, receiver, interferometer, sensitivity, signal-to-noise ratio, gyroscope, integration, extended Kalman filter

## Abstract

Attitude determination systems based on Global Navigation Satellite Systems (GNSS) work on principle of phase interferometer, using multiple receiving antennas. They rely on a good quality of carrier phase tracking, that is not the case in real dynamic environment with low signal-to-noise ratio (SNR), for example, in a ground vehicle moving through an urban area or forest. There is still a problem in providing a GNSS attitude in such common conditions. This research is focused on improving sensitivity (i.e., the capability of providing attitude at a low SNR) and the reliability of the GNSS attitude determination system. It is contrasted with the majority of publications, where precision or computational efficiency is the main goal, but sensitivity and reliability are out of their scope. In the proposed system, sensitivity improved by using two measures: (a) tracking only phase differences instead of tracking full carrier phases—this is more sensitive due to the lower dynamics of the underlying process, and (b) using deep integration with gyroscope, where all phase differences are tracked in a vector gyro-aided loop closed on user’s attitude in state vector. The algorithm synthesis is given, and simulation results are presented in this article. This shows that the minimal working SNR is lowered from 27–36 dBHz (typical) down to 20 dBHz, even with a low-cost MEMS gyroscope.

## 1. Introduction

Navigation systems are intended for the determination of coordinates, velocity and spatial orientation (attitude) that is described by rotation angles between body frame and navigation frame. For attitude determination, there are widely used inertial sensors (accelerometers and gyroscopes) comprising inertial navigation systems (INS) [1,2]. In recent decades, with the growing popularity of global navigation satellite systems (GNSS), [3,4,5] are commonly used GNSS-based attitude determination systems working on the principle of radio interferometer. The advantages of inertial sensors are autonomy and absolute immunity to interferences. Their drawbacks are errors increasing with time, the need for initial alignment and the very high cost for reasonable precision. The advantages of GNSS are a relatively low and stable attitude estimation error and good cost. On the other hand, GNSS’ attitude estimation error highly depends on receiving conditions, such as signal-to-noise ratio (SNR) and the presence of multipath and interference, and also on user dynamics and the distance between receiving antennas (baseline). One of the approaches to upgrading attitude determination system efficiency is developing the inertial-satellite-navigation system (ISNS), in which signals of inertial sensors and GNSS are jointly processed. The efficiency growth in ISNS is due to the fact that INS and GNSS are mutually complementary, i.e., they work on different physical principles [6]. The integrated GNSS/INS attitude determination system potentially possesses higher accuracy, sensitivity, continuity, reliability and anti-jam capability in comparison to separate attitude sensors. A lot of corresponding publications are devoted to the problem of the integrated estimation of linear movement parameters: coordinates and velocity [7,8,9,10,11]. The commonly used approaches to the integrated processing of INS and GNSS signals [6] are: loosely coupled [11], tightly coupled [10], deeply integrated [7,9]. The best characteristics of accuracy and noise/interference immunity are achieved in ISNSs with deeply integrated processing algorithms.

There are also integrated GNSS/INS processing algorithms for problems of attitude determination [12,13,14,15,16]. The most notable feature of these algorithms are the exploitation of several (2–4) GNSS antennas. The loosely coupled integration approach is prevalent: the inputs are the raw carrier phase and pseudo-range measurements from the GNSS receiver and the measurements of angular rate and acceleration vectors from three-axis gyroscopes and accelerometers [15]. The information about orientation is contained in the phase differences in GNSS signals received by spaced antennas. Therefore, in some way or another, the mentioned phase differences are used in the integration algorithm as the primal source of attitude information. It is known [15,17] that phase measurements are ambiguous. For phase ambiguity resolution, there are appropriate algorithms, for instance, the Lambda method [18]. It is necessary to note that phase is measured in the GNSS receiver with a phase-locked loop (PLL). PLL is the loop that is least immune to the noise and interference, and incapable of tracking weak signals with an SNR below 27 dBHz (typically). For assured ambiguity resolution phase-tracking, RMS error must not be more than the 2°–3° that is achieved only at SNR around 36 dBHz. Actual SNR can be much less in a real dynamic environment for ground vehicles moving through an urban area or just forest. There is still problem to provide GNSS attitude in such common conditions. The sensitivity (i.e., capability of providing attitude at a low SNR) of the GNSS/INS attitude determination system must be improved.

One of the solutions of this problem was proposed in [19]; it is based on the direct tracking of phase differences for the same signal received by different antennas. Instead of a coherent tracking of the signal’s carrier phase, only Doppler and delay are tracked with non-coherent loops that do not require phase lock and provide a 6–8 dB greater sensitivity than PLL. In this article, we enhance this approach by adding measurements from the three-axis gyro. These measurements, along with the GNSS signals from three antennas, are processed in an integration algorithm which evolved with a deeply integrated approach. The angular rate information from gyro helps in tracking phase differences by canceling angular dynamics. The phase differences for all visible signals are continuously tracked in a vector loop. Vector dynamic aiding improves the sensitivity and interference immunity of the whole attitude determination system. This approach is contrasted to the ones in the known literature, where precision or ambiguity resolution efficiency is the main goal, but sensitivity and reliability are out of the scope. Vector loop and the deep integration with a gyroscope eliminate the need for ambiguity resolution, as well as cycle slip detection and repairs in runtime. Only at initialization and after a long GNSS outage is ambiguity resolution required.

It is known [2] that orientation can be described by Euler’s angles, transformation matrix, rotation vector, quaternions, Rodrigues parameters, etc. In [20], we described the Euler’s angle-based, deeply integrated attitude determination algorithm for ISNS. In recent years, quaternion representation is more often used, and, in this article, it will be used as well.

In the majority of publications (including [1,8,9,10,11,12,13,14,15,16]), accelerometer and gyro outputs are considered as measurements. In this article, a different method is used: the gyro measurements are substituted into the state vector (quaternion) dynamic model as a known function, as we described in [6,21]. This approach, on the one hand, simplifies an integration algorithm computationally and, on the other hand, ensures smooth dead-reckoning functionality when GNSS signals are lost. The gyro errors (biases, scales, misalignments) are included in the state vector, so they are estimated when GNSS signals are strong and compensated for during GNSS outage.

Another feature of the proposed attitude estimation algorithm is the algorithm synthesis method. In [12,13,14,15,16], the Wahba quadratic form [22] is used for the synthesis. In this article, statistical nonlinear filtering theory [23,24] is used, and, particularly, the extended Kalman filter (EKF) is taken as the base for integration algorithm. For synthesis purposes, the original model of received signals’ carrier phases is used, which consists of two components caused by the transitional and rotational movement of the antenna system. This could synthesize the algorithm for the direct tracking of inter-antenna carrier phase differences that include rotational movement only.

The proposed deeply integrated GNSS/gyro attitude determination algorithm is recommended for implementation in navigational equipment that works in harsh receiving conditions with a low SNR, multipath, shadowing and high user dynamics in urban canyons or forest environment.

## 2. Algorithm Synthesis

For 3D attitude determination, three spaced GNSS antennas (as a minimum required number) are used, and a triad of gyroscopes. Attitude estimation algorithm is synthesized by using the statistical theory of optimal nonlinear filtering [23,25]. For this purpose, we shall formulate the statistical description of:I/Q components at the GNSS receiver’s correlator output, considered as the observations containing information about attitude hidden in carrier phase differences between antennas;Gyro measurements;Estimated state vector’s dynamic model.

The special model is used for GNSS signal’s carrier phase observed at spaced antennas. In this model, carrier phase is presented as the sum of two terms: one is defined by the transitional movement of the antenna plane along the user-satellite direction, and the other is defined by the rotational movement of the antennas plane around the chosen center of the body. This representation can exclude “transitional” terms from further synthesis, and process phase differences consisting of “rotational” terms only.

For the statistical description of gyro measurements, the random walk models are used for the sensor errors: biases, scales, misalignments. The attitude will be handled in quaternion form.

The important feature of the algorithm under development is the choice of estimated state vector. In state vector, we include quaternion representing user’s attitude, gyroscope biases, scale factors and axes misalignments, and unknown phase biases in radio-frequency front-ends. The stochastic dynamic model for this state vector is developed using gyro measurements as a known function.

The deep integration approach is used for algorithm synthesis. According to this approach, optimal estimation system can be considered as vector-tracking loop consisting of a vector discriminator (where all the received GNSS signals are processed) and integration filter (that processes discriminator outputs and gyroscope measurements). Discriminator equations’ derivation implies the cancellation of “transitional” terms of carrier phases. Error signals at discriminator outputs are proportional to estimation errors in inter-antenna phase differences that include only “rotational” terms of carrier phases. Vector feedback signal was formed out of predicted values of inter-antenna phase differences and corrected with estimated front-end phase biases. These values are calculated from a predicted estimate of attitude quaternion inside the integration filter. The estimate of quaternion at the output of the integration filter then transformed to estimates of conventional roll, pitch and yaw angles.

### 2.1. Coordinate Systems and Phase Differences

Considering the Earth-Centered Earth-Fixed reference frame OXYZ (Figure 1) as a navigation frame, it will be denoted as ecef. Moving object has a dedicated body reference frame $O_{c}X_{c}Y_{c}Z_{c}$ with origin $O_{c}$, also denoted as rpy. The locations of GNSS antenna phase centers are denoted as $A_{j}$, $j=\bar{1,3}$, their coordinates are considered to be known in rpy frame. Lines $A_{1}A_{2}$, $A_{1}A_{3}$, $A_{2}A_{3}$ are named as baselines.

Assuming that the gyro triad is located at the point $O_{c}$, its axes are aligned to rpy frame. Therefore, the gyro measures angular rate vector in rpy frame.

The direction of signal arrival from i-th navigation satellite (NS) ($i=\bar{1,n}$) will be described by vector $\mu_{\mathrm{NS},i}$, specified in ecef frame
(1)$$\mu_{\mathrm{NS},i}={\left|\begin{array}{ccc} \frac{x_{\mathrm{NS},i}-x_{O_{c}}}{R_{i}} & \frac{y_{\mathrm{NS},i}-y_{O_{c}}}{R_{i}} & \frac{z_{\mathrm{NS},i}-z_{O_{c}}}{R_{i}} \end{array}\right|}^{T},$$
where $x_{O_{c}}$, $y_{O_{c}}$, $z_{O_{c}}$ are coordinates of the point $O_{c}$ in ecef frame; $R_{i}$ is a distance to i-th NS; $x_{\mathrm{NS},i}$, $y_{\mathrm{NS},i}$, $z_{\mathrm{NS},i}$ are coordinates of i-th NS in ecef frame.

Then, three vectors are introduced $L_{A_{j}O_{c}}^{rpy}$, $j=\bar{1,3}$ that contain coordinates of points $A_{j}$ in rpy frame. The transformation of vectors $L_{A_{j}O_{c}}^{rpy}$, $j=\bar{1,3}$ from rpy frame to ecef frame is given by the equation
(2)$$L_{A_{j}O_{c}}^{ecef}\left(\alpha \right)=C_{rpy}^{ecef}\left(\alpha \right)L_{A_{j}O_{c}}^{rpy},\;j=\bar{1,3},$$
where $\alpha ={\left|\begin{array}{ccc} R & P & Y \end{array}\right|}^{T}$; R, P, Y are the roll pitch and yaw angles respectively; $C_{rpy}^{ecef}$ is the coordinate transformation matrix from rpy to ecef frame.

GNSS signals observed at the points $A_{j}$, $j=\bar{1,3}$ are phase-shifted with respect to the signal observed at the point $O_{c}$. The phase difference of each signal, relatively, $O_{c}$ is
(3)$$\psi_{A_{j}}^{i}\left(\alpha \right)=\frac{2\pi \mu_{\mathrm{NS},i}^{T}L_{A_{j}O_{c}}^{ecef}\left(\alpha \right)}{\lambda },\;j=\bar{1,3},$$
where λ is the signal’s wavelength. By substituting Equation (2) to (3), we can describe phase difference in a more straight-forward form keeping in mind that $L_{A_{j}O_{c}}^{rpy}$ is a known constant vector:(4)$$\psi_{A_{j}}^{i}\left(\alpha \right)=\frac{2\pi }{\lambda }\mu_{\mathrm{NS},i}^{T}C_{rpy}^{ecef}\left(\alpha \right)L_{A_{j}O_{c}}^{rpy},\;j=\bar{1,3},$$

From (3) and (4), it is obvious that information about attitude α is contained in relative phases $\psi_{A_{j}}^{i}\left(\alpha \right)$, $j=\bar{1,3}$ of signals received by antennas $A_{j}$. Hence, the general principle of attitude determination using GNSS signals is based on the observation of relative phases $\psi_{A_{j}}^{i}$, $j=\bar{1,3}$, $i=\bar{1,n}$ in GNSS receiver and estimation attitude by using functional dependency Equation (3) in some way. The extended Kalman filer is used here for this purpose.

### 2.2. Gyro Measurements and Attitude Integration

Angular rate measurements at the output of three-axis strapdown gyroscope can be modeled as
(5)$$y_{\omega_{rpy};k}=\left(I+m_{g;k}\right)\Omega_{rpy;k}+b_{g;k}+n_{g;k}$$
where $\Omega_{rpy;k}={\left|\begin{array}{ccc} \omega_{r,k} & \omega_{p,k} & \omega_{y,k} \end{array}\right|}^{T}$ is the true angular rate vector in *rpy* frame; $m_{g;k}$ is the matrix of axes misalignments and scale factor errors; $b_{g;k}$ is the bias vector; $n_{g;k}$ is a vector of mutually independent discrete zero-mean white Gaussian noises with variances $\sigma_{g}^{2}$.

The user’s attitude will be defined by quaternion $q_{rpy}^{ecef}$ representing a rotation of the rpy frame axes to the *ecef* frame axes. We can rewrite phase difference (4) as the function of this quaternion
(6)$$\psi_{A_{j}}^{i}\left(q_{rpy}^{ecef}\right)=\frac{2\pi }{\lambda }\mu_{\mathrm{NS},i}^{T}\left(C\left(q_{rpy}^{ecef}\right)L_{A_{j}O_{c}}^{rpy}\right),\;j=\bar{1,3},$$
where $C\left(q_{rpy}^{ecef}\right)≜C_{rpy}^{ecef}\left(\alpha \right)$.

The evolution of the quaternion $q_{rpy}^{ecef}$ can be described by the integration of angular rate vector $\Omega_{rpy;k}$ and earth rotation rate with the following equation
(7)$$q_{rpy;k}^{ecef}=\Delta_{E}^{*}⊗q_{rpy;k-1}^{ecef}⊗\Delta_{rpy;k}^{}\left(\Omega_{rpy}\left(t\right)\right),$$
where $\Delta_{E}^{*}$ is the constant quaternion representing rotation of Earth over the time interval $t_{k-1}\ldots t_{k}$; $\Delta_{rpy;k}^{}\left(\Omega_{rpy}\left(t\right)\right)$ is the update quaternion representing the *rpy* frame rotation in inertial space over time interval $t_{k-1}\ldots t_{k}$; ⊗—is the quaternion product.

### 2.3. Correlation Signal Processing in GNSS Receiver

All modern GNSS receivers are contain a multi-channel correlator that carries out the multiplication and accumulation of the input signal with the quadrature reference signal as follows
(8)$$I_{j;k}^{i}=\sum_{l=1}^{M}y_{A_{j};k,l}\cdot s_{ref,j,I}^{i}\left(t_{k,l}\right),\;Q_{j;k}^{i}=\sum_{l=1}^{M}y_{A_{j};k,l}\cdot s_{ref,j,Q}^{i}\left(t_{k,l}\right),$$
where $s_{ref,j,I}^{i}\left(t_{k,l}\right)$, $s_{ref,j,Q}^{i}\left(t_{k,l}\right)$ are the quadrature components of reference signal; $y_{A_{j};k,l}$ is the input signal from antenna $A_{j}$, amplified and down-converted in corresponding front-end; k,l are the time indexes corresponding to the double-indexed time scale $t_{k,l}$ shown in Figure 2.

In this diagram, the second index of time moments $t_{k,l}$, $t_{k,l+1}$, $t_{k,l+2}$ …, $l=\bar{1,M}$, corresponds to the time step $T_{d}$—the step of fast signal processing after analog-to-digital converter, and the first index of time moments $t_{k-1}=t_{k-1,0}$, $t_{k}=t_{k,0}$, $t_{k+1}=t_{k+1,0}$ … corresponds to slow processing with a time step $T=MT_{d}$. Rapidly changing signal waveforms $y_{A_{j};k,l}$ need to be described using “fast” samples at the moments $t_{k,l}$ within interval $\left[t_{k},t_{k+1}\right]$. A description of slowly changing processes, such as correlator I/Q outputs, time delays, phases, Doppler shifts and orientation, is given at moments $t_{k}$. We also assume that angular rate measurements from gyro are synchronized with moments $t_{k}$.

When the GNSS receiver is locked, the *i*–th NS signal, the delay tracking error and frequency tracking error can be considered small. We suppose that the amplitudes of received *i*–th NS signal at different antennas are equal $A_{j}^{i}=A^{i}$, $j=\bar{1,3}$. In this case, the correlator outputs can be expressed as [6]
(9)$$I_{j;k}^{i}=\frac{A^{i}M}{2}\cos\left(\varphi_{0;k}^{i}+\pi \vartheta_{\mathrm{DI};k}^{i}+\psi_{A_{j};k}^{i}+\delta_{A_{j}}+\frac{T}{2}\left(\omega_{D,k}^{i}+\omega_{A_{j},\psi ;k}^{i}-{\tilde{\omega }}_{D,k}^{i}\right)\right)+n_{I,j;k}^{i},\;Q_{j;k}^{i}=\frac{A^{i}M}{2}\sin\left(\varphi_{0;k}^{i}+\pi \vartheta_{\mathrm{DI};k}^{i}+\psi_{A_{j};k}^{i}+\delta_{A_{j}}+\frac{T}{2}\left(\omega_{D,k}^{i}+\omega_{A_{j},\psi ;k}^{i}-{\tilde{\omega }}_{D,k}^{i}\right)\right)+n_{Q,j;k}^{i},$$
where $\varphi_{0;k}^{i}$ is the carrier phase tracking error at the beginning of interval $\left[t_{k},t_{k+1}\right]$ with respect to the point $O_{c}$; $\vartheta_{\mathrm{DI};k}^{i}$ is the bit of digital information transmitted within navigation signal that can be 0 or 1; $\psi_{A_{j};k}^{i}$ is the carrier phase shift (3) at the beginning of interval $\left[t_{k},t_{k+1}\right]$; $\delta_{A_{j}}$ is the phase bias of the j-th antenna front-end; $\omega_{D,k}^{i}$ is a Doppler frequency shift due to motion of point $O_{c}$ relative to i-th NS, $\omega_{A_{j},\psi ;k}^{i}$ is a Doppler shift in the signal received at the point $A_{j}$ due to object rotation relative to the point $O_{c}$ (we suppose $\omega_{D,k}^{i}=const$, $\omega_{A_{j},\psi ;k}^{i}=const$ over time interval $\left[t_{k},t_{k+1}\right]$); ${\tilde{\omega }}_{D,k}^{i}$ is the Doppler frequency shift in the reference signal; $n_{I,j;k}^{i}$, $n_{Q,j;k}^{i}$ are non-correlated zero-mean white Gaussian noises with variances $D_{IQ}=\frac{1}{2q_{c/n_{0},i}T}{\left(\frac{A^{i}M}{2}\right)}^{2}$, $q_{c/n_{0},i}=\frac{{\left(A^{i}\right)}^{2}}{2N_{0}}$. We have taken into account that, on the interval $\left[t_{k},t_{k+1}\right]$ carrier phase and phase difference, $\psi_{A_{j};k,l}^{i}$ (4) are changed linearly, i.e., $\psi_{A_{j};k,l}^{i}=\psi_{A_{j};k}^{i}+\left(l-1\right)T_{d}\omega_{A_{j},\psi ;k}^{i}$.

There is an inevitable signal delay in the GNSS receiver’s signal chain (front-end) from the antenna to ADC. Because of technological differences, these delays will be different for different front-ends. Therefore, we have taken into account this phase lag as an add-on $\delta_{A_{j}}^{}$ to the signal phase in front-end *j*. Supposing the use of Code Division Multiple Access (CDMA) signals only, an add-on $\delta_{A_{j}}^{}$ will be stated as equal for all navigation signals received on the antenna $A_{j}$.

Quadrature signals $I_{j;k}^{i}$, $Q_{j,k}^{i}$ can be expressed in a complex form ${\dot{X}}_{j;k}^{i}=I_{j;k}^{i}+jQ_{j;k}^{i}$ where $j≜\sqrt{-1}$ is an imaginary unit. For further simplification, but without loss of generality, the origin $O_{c}$ of *rpy* frame is put to the phase center of the antenna $A_{1}$. Then, correlator outputs for *i*–th NS signal and the j-th antenna ($j=\bar{1,3}$) can be rewritten in the form
(10)$${\dot{X}}_{1;k}^{i}=\frac{A^{i}M}{2}\exp\left\{j\cdot \left(\varphi_{0;k}^{i}+\pi \vartheta_{\mathrm{DI};k}^{i}+\delta_{A_{1}}+\frac{T}{2}\left(\omega_{D,k}^{i}-{\tilde{\omega }}_{D,k}^{i}\right)\right)\right\}+{\dot{n}}_{X,1;k}^{i},\;{\dot{X}}_{2;k}^{i}=\frac{A^{i}M}{2}\exp\left\{j\cdot \left(\varphi_{0;k}^{i}+\pi \vartheta_{\mathrm{DI};k}^{i}+\psi_{A_{2};k}^{i}+\delta_{A_{2}}+\frac{T}{2}\left(\omega_{D,k}^{i}-{\tilde{\omega }}_{D,k}^{i}+\omega_{A_{2},\psi ;k}^{i}\right)\right)\right\}+{\dot{n}}_{X,2;k}^{i},\;{\dot{X}}_{3;k}^{i}=\frac{A^{i}M}{2}\exp\left\{j\cdot \left(\varphi_{0;k}^{i}+\pi \vartheta_{\mathrm{DI};k}^{i}+\psi_{A_{3};k}^{i}+\delta_{A_{3}}+\frac{T}{2}\left(\omega_{D,k}^{i}-{\tilde{\omega }}_{D,k}^{i}+\omega_{A_{3},\psi ;k}^{i}\right)\right)\right\}+{\dot{n}}_{X,3;k}^{i},$$
where ${\dot{n}}_{X,j;k}^{i}=n_{I,j;k}^{i}+j\cdot n_{Q,j;k}^{i}$.

In [6,19], it is shown that, even with a non-coherent tracking of the *i*–th NS signal, there can be computed estimations of phase differences $\psi_{A_{j};k}^{i}$, $j=\bar{2,3}$ by using products
(11)$${\dot{Y}}_{2;k}^{i}={\dot{X}}_{2;k}^{i}\cdot {\left({\dot{X}}_{1;k}^{i}\right)}^{*}=\frac{{\left(A^{i}M\right)}^{2}}{2}\exp\left\{j\cdot \left(\psi_{A_{2};k}^{i}+\delta_{A_{2}/A_{1}}+T\omega_{A_{2},\psi ;k}^{i}/2\right)\right\}+{\dot{n}}_{Y,2;k}^{i},\;{\dot{Y}}_{3;k}^{i}={\dot{X}}_{3;k}^{i}\cdot {\left({\dot{X}}_{1;k}^{i}\right)}^{*}=\frac{{\left(A^{i}M\right)}^{2}}{2}\exp\left\{j\cdot \left(\psi_{A_{3};k}^{i}+\delta_{A_{3}/A_{1}}+T\omega_{A_{3},\psi ;k}^{i}/2\right)\right\}+{\dot{n}}_{Y,3;k}^{i}$$
where $\delta_{A_{2}/A_{1}}=\delta_{A_{2}}-\delta_{A_{1}}$, $\delta_{A_{3}/A_{1}}=\delta_{A_{3}}-\delta_{A_{1}}$; ${\dot{n}}_{Y,2;k}^{i}$, ${\dot{n}}_{Y,3;k}^{i}$—correlated complex zero-mean white Gaussian noises with variances $D_{Y}=D_{IQ}\left(2{\left(A^{i}M/2\right)}^{2}+D_{IQ}\right)$ and the covariance $R_{n_{Y2/Y3}}=D_{IQ}{\left(A^{i}M/2\right)}^{2}\exp\left(j\cdot \left(\delta_{A_{2}}-\delta_{A_{3}}+T\left(\omega_{A_{2},\psi ;k}^{i}-\omega_{A_{3},\psi ;k}^{i}\right)/2\right)\right)$.

Complex signals (11) can be considered as the measurements of phase differences $\psi_{A_{j},\psi ;k}^{i}$, $j=\bar{2,3}$ in which the “transitional” part of the carrier phase tracking error is compensated along with the term responsible for the reference oscillator instability that is common for all three antennas. Still, there remain unknowns, $\delta_{A_{2}/A_{1}}$ and $\delta_{A_{3}/A_{1}}$, caused by non-identical front-end phase biases.

### 2.4. Synthesis of Deeply Integrated GNSS/Gyro Attitude Determination Algorithm

Taking into account that $\psi_{A_{2};k}^{i}$, $\psi_{A_{3};k}^{i}$ in (11) are functions of the quaternion $q_{rpy,k}^{ecef}$ (6), let us define the state vector to be estimated as $x_{k}={\left|{\left(q_{rpy,k}^{ecef}\right)}^{T}\;b_{g,k}^{T}\;S_{g,k}^{T}{\vec{m}}_{g,k}^{T}\;\delta_{k}^{T}\right|}^{T}$,

where $S_{g}={\left|m_{g,11}\;m_{g,22}\;m_{g,33}\right|}^{T}$—the diagonal elements of the matrix $m_{g}$ represented as a vector; ${\vec{m}}_{g}={\left|m_{g,12}\;m_{g,13}\;m_{g,23}\right|}^{T}$—three off-diagonal elements of the matrix $m_{g}$ represented as a vector; $\delta_{k}={\left|\begin{array}{cc} \delta_{A_{2}/A_{1}} & \delta_{A_{3}/A_{1}} \end{array}\right|}^{T}$.

A dynamic model of quaternion $q_{rpy,k}^{ecef}$ is given by (7), where the Earth rotation quaternion $\Delta_{E}^{*}$ and update quaternion $\Delta_{rpy;k}^{}$ are described by formulas
(12)$$\Delta_{RPY;k}^{}=\left|\begin{array}{c} \cos\left(\left\|\rho_{k}\right\|/2\right)\\ \frac{\rho_{1;k}}{\left\|\rho_{k}\right\|}\sin\left(\left\|\rho_{k}\right\|/2\right)\\ \frac{\rho_{2;k}}{\left\|\rho_{k}\right\|}\sin\left(\left\|\rho_{k}\right\|/2\right)\\ \frac{\rho_{3;k}}{\left\|\rho_{k}\right\|}\sin\left(\left\|\rho_{k}\right\|/2\right) \end{array}\right|,\;\Delta_{E}^{*}=\left|\begin{array}{l} \cos\left(\frac{\omega_{E}T}{2}\right)\\ 0\\ 0\\ -\sin\left(\frac{\omega_{E}T}{2}\right) \end{array}\right|,$$
where $\omega_{E}$ is the Earth rotation rate; $\rho_{k}$ is the rotation vector representing a rotation of the rpy frame in inertial space over the time interval $t_{k-1}\ldots t_{k}$. If this interval is small, $\rho_{k}$ can be approximately computed by a numerical integration of angular rate $\Omega_{rpy}\left(t\right)$
(13)$$\rho_{k}\approx \frac{T}{2}\left(\Omega_{rpy,k}+\Omega_{rpy,k-1}\right)=T\left(I-m_{g;k-1}\right)\left[\frac{1}{2}\left(y_{\omega_{rpy};k}+y_{\omega_{rpy};k-1}\right)-b_{g;k-1}\right]+\xi_{\rho ;k},$$
where $\xi_{\rho ;k}\approx -\frac{T}{2}\left(n_{g;k}+n_{g;k-1}\right)$.

For errors ${\vec{m}}_{g,k}$ and $b_{g,k}$ we use a stochastic model in the form of Wiener process (random walk)
(14)$${\vec{m}}_{g,k}={\vec{m}}_{g,k-1}+\xi_{\vec{m}g,k},\;b_{g,k}=b_{g,k-1}+\xi_{bg,k}$$
where $\xi_{bg,k}$ is a vector of three independent white Gaussian noises with zero expectations and variances $\sigma_{bg}^{2}$; $\xi_{\vec{m}g,k}$ is a vector of three independent white Gaussian noises with zero expectations and variances $\sigma_{mg}^{2}$.

A similar model is used for the front-end phase biases
(15)$$\delta_{k}=\delta_{k-1}+\xi_{\delta ,k}$$
where $\xi_{\delta ,k}$ is a vector of independent white Gaussian noises with zero expectations and variances $\sigma_{\delta }^{2}$.

Equations (7), (12)–(15) describe a Markov model for the state vector $x_{k}$ that can be represented in the general vector form as
(16)$$x_{k}=F\left(x_{k-1},y_{\omega_{rpy};k},y_{\omega_{rpy};k-1},\xi_{x;k}\right)$$
where $\xi_{x;k}={\left|\begin{array}{ccc} \frac{1}{2}{\left(n_{g;k}^{}+n_{g;k-1}^{}\right)}^{T} & \begin{array}{cc} \xi_{bg;k}^{T} & \xi_{S_{g};k}^{T} \end{array} & \begin{array}{cc} \xi_{{\vec{m}}_{g};k}^{T} & \xi_{\delta ,k}^{T} \end{array} \end{array}\right|}^{T}$ is a vector of 14 independent zero-mean white Gaussian noises with variances: $\sigma_{g}^{2}$ (three components), $\sigma_{bg}^{2}$ (three components), $\sigma_{mg}^{2}$ (six components), $\sigma_{\delta }^{2}$ (two components); $F\left(x_{k-1},y_{\omega_{rpy};k},y_{\omega_{rpy};k-1},\xi_{x;k}\right)$ is a nonlinear function determined by the right sides of Equations (9), (12)–(15).

It is necessary to note that gyro output $y_{\omega_{rpy};k}$ is considered to be a known function of time in the context of state dynamics (16).

According to control theory, each tracking loop must contain a discriminator and filter. Complex signals (11) can be used to obtain discriminators for biased phase differences $\psi_{A_{j};k}^{i}+\delta_{A_{j}/A_{1}}$, $j=\bar{2,3}$
(17)uDiscr,ψAj;ki=Im(Y˙j;ki·exp{−j·(ψ˜Aj;ki+δ˜Aj/A1+Tω˜Aj,ψ;ki/2)})
where ${\tilde{\psi }}_{A_{j};k}^{i}$ is the predicted (in filter) estimate of phase difference, ${\tilde{\delta }}_{A_{j}/A_{1}}$ is the predicted estimate of front-end *j* phase bias (relative to front-end 1), ${\tilde{\omega }}_{A_{j},\psi ;k}^{i}$ is the predicted estimate of phase difference rate that can be computed approximately by numerical derivation of Equation (6) with substitution $q_{rpy,k}^{ecef}\to {\tilde{q}}_{rpy,k}^{ecef}$. Hence the computation algorithm for ${\tilde{\omega }}_{A_{j},\psi ;k}^{i}$ is following: ${\tilde{\omega }}_{A_{j},\psi ;k}^{i}\approx \frac{2\pi }{T\lambda }\mu_{\mathrm{NS},i}^{T}\left(C\left({\tilde{q}}_{rpy,k}^{ecef}\right)-C\left({\tilde{q}}_{rpy,k-1}^{ecef}\right)\right)\left(L_{A_{j}O_{c}}^{rpy}-L_{A_{1}O_{c}}^{rpy}\right)$.

In [20], it is shown that, if the tracking error $\Delta_{\psi }=\psi_{A_{j};k}^{i}-{\tilde{\psi }}_{A_{j};k}^{i}$ is small, the phase difference discriminator can be linearized to form
(18)$$u_{Discr,\psi_{A_{j}};k}^{i}=S_{discr}^{i}\left(\psi_{A_{j};k}^{i}-{\tilde{\psi }}_{A_{j};k}^{i}+\delta_{A_{j}/A_{1}}-{\tilde{\delta }}_{A_{j}/A_{1}}\right)+n_{j,k}^{i},\;i=\bar{1,n},\;j=\bar{2,3},$$
where $S_{discr}^{i}={\left(A^{i}M\right)}^{2}/2$ is the slope of the discrimination function; $n_{j,k}^{i}$ is the noise at the discriminator output.

Using the methodology described in [6], equivalent linear observations are introduced
(19)$${\tilde{y}}_{\psi_{A_{j}};k}^{i}=S_{discr}^{i}\left(\psi_{A_{j};k}^{i}+\delta_{A_{j}/A_{1}}\right)+n_{j,k}^{i},\;i=\bar{1,n},\;j=\bar{2,3}.$$

The observations (19) are combined in a vector with the following order ${\tilde{y}}_{\psi ,k}={\left|\begin{array}{cccc} {\tilde{y}}_{\psi_{A_{2}};k}^{1}, & {\tilde{y}}_{\psi_{A_{3}};k}^{1} & \ldots & \begin{array}{cc} {\tilde{y}}_{\psi_{A_{2}};k}^{n}, & {\tilde{y}}_{\psi_{A_{3}};k}^{n} \end{array} \end{array}\right|}^{T}$.

For the synthesis of the integration filter, we use equivalent linear observations ${\tilde{y}}_{\psi ,k}$, where parameters $\psi_{A_{j};k}^{i}$, $\delta_{A_{j}/A_{1}}$ are the functions of the state vector $x_{k}$, i.e.,
(20)$${\tilde{y}}_{\psi ,k}=S_{\psi }\left({\tilde{x}}_{k}\right)+n_{k},$$
(21)$$S_{\psi }\left({\tilde{x}}_{k}\right)=\left|\begin{array}{c} S_{discr}^{1}\left({\tilde{\psi }}_{A_{2};k}^{1}+{\tilde{\delta }}_{A_{2}/A_{1}}\right)\\ S_{discr}^{1}\left({\tilde{\psi }}_{A_{3};k}^{1}+{\tilde{\delta }}_{A_{3}/A_{1}}\right)\\ \ldots \\ S_{discr}^{n}\left({\tilde{\psi }}_{A_{2};k}^{n}+{\tilde{\delta }}_{A_{2}/A_{1}}\right)\\ S_{discr}^{n}\left({\tilde{\psi }}_{A_{3};k}^{n}+{\tilde{\delta }}_{A_{3}/A_{1}}\right) \end{array}\right|,\;{\tilde{\psi }}_{A_{j};k}^{i}=\frac{2\pi }{\lambda }\mu_{\mathrm{NS},i}^{T}C\left({\tilde{q}}_{rpy,k}^{ecef}\right)\left(L_{A_{j}O_{c}}^{rpy}-L_{A_{1}O_{c}}^{rpy}\right).$$
Here, $S_{\psi }\left({\tilde{x}}_{k}\right)$ is the observations vector function, and $n_{k}={\left|\begin{array}{ccc} n_{2,k}^{1},n_{3,k}^{1} & \ldots & n_{2,k}^{n},n_{3,k}^{n} \end{array}\right|}^{T}$.

A vector discriminator of the following form can be associated with the observations (20)
(22)$$u_{Discr,\psi ;k}={\left|u_{Discr,\psi_{A_{2}};k}^{1},\;u_{Discr,\psi_{A_{3}};k}^{1}\;\ldots \;u_{Discr,\psi_{A_{2}};k}^{n},u_{Discr,\psi_{A3};k}^{n}\right|}^{T}.$$

Now, the observations vector (20) and the state vector $x_{k}$, described by the vector Equation (16), can be substituted in general state estimation equations of the extended Kalman filter [24]
(23)$${\hat{x}}_{k}={\tilde{x}}_{k}+K_{k}\left({\tilde{y}}_{\psi ,k}-S_{\psi }\left({\tilde{x}}_{k}\right)\right),\;{\tilde{x}}_{k}=f\left({\hat{x}}_{k-1}\right),\;K_{k}={\tilde{E}}_{k}H_{k}^{T}{\left(H_{k}{\tilde{E}}_{k}H_{k}^{T}+D_{n_{\psi }}\right)}^{-1}$$
(24)$${\tilde{E}}_{k}=\frac{\partial f\left({\hat{x}}_{k-1}\right)}{\partial x}E_{k-1}{\left(\frac{\partial f\left({\hat{x}}_{k-1}\right)}{\partial x}\right)}^{T},\;E_{k}=\left(I-K_{k}H_{k}\right){\tilde{E}}_{k}$$
where $K_{k}$ is a Kalman gain matrix; $E_{k}$ is the a posteriori matrix of estimation covariance; ${\tilde{E}}_{k}$ is prediction covariance matrix for the predicted value of the estimated state vector ${\tilde{x}}_{k}$; $D_{\xi_{x},k}$ is the covariance matrix of dynamic disturbance noise (given in Appendix A);
(25)$$H_{k}=\frac{\partial S_{\psi }\left({\tilde{x}}_{k}\right)}{\partial x}$$
measurement sensitivity (or observation) matrix;
$$f\left({\hat{x}}_{k-1}\right)\equiv F\left({\hat{x}}_{k-1},y_{\omega_{rpy};k},y_{\omega_{rpy};k-1},\xi_{x;k}=0\right).$$

The formulas for the derivative matrices $\frac{\partial f\left({\hat{x}}_{k-1}\right)}{\partial x}$ and $\frac{\partial S_{\psi }\left({\tilde{x}}_{k}\right)}{\partial x}$ that are present in Equations (24) and (25) are given in Appendix A.

Taking into account Equations (17) and (18), Equation (23) can be rewritten in the form
(26)$${\hat{x}}_{k}={\tilde{x}}_{k}+K_{k}u_{Discr,\psi ;k}.$$

Finally, the proposed algorithm for attitude determination with GNSS signals and the three-axis gyro is described by Equations (11)–(13), (17), (21), (24)–(26).

The output attitude is represented in the form of quaternion $q_{rpy,k}^{ecef}$. To obtain conventional Euler angles of roll, pitch and yaw, we have to make the following calculations
(27)$$U=C_{rpy}^{ned}\left(\alpha \right)={\left(C_{ned}^{ecef}\right)}^{T}C\left(q_{rpy,k}^{ecef}\right),\;R=\mathrm{atan}2\left(U_{3,2},\;U_{3,3}\right),\;P=\mathrm{atan}2\left(\sqrt{{\left(U_{1,1}\right)}^{2}+{\left(U_{2,1}\right)}^{2}},\;U_{3,1}\right)-\frac{\pi }{2},\;Y=\mathrm{atan}2\left(U_{2,1},\;U_{1,1}\right),$$
where
(28)$$C\left(q\right)=\left|\begin{array}{ccc} q_{1}^{2}+q_{2}^{2}-q_{3}^{2}-q_{4}^{2} & 2\left(q_{2}q_{3}-q_{1}q_{4}\right) & 2\left(q_{1}q_{3}+q_{2}q_{4}\right)\\ 2\left(q_{2}q_{3}+q_{1}q_{4}\right) & q_{1}^{2}+q_{3}^{2}-q_{2}^{2}-q_{4}^{2} & 2\left(q_{3}q_{4}-q_{1}q_{2}\right)\\ 2\left(q_{2}q_{4}-q_{1}q_{3}\right) & 2\left(q_{1}q_{2}+q_{3}q_{4}\right) & q_{1}^{2}+q_{4}^{2}-q_{3}^{2}-q_{2}^{2} \end{array}\right|,$$
and
(29)$$C_{ned}^{ecef}=\left|\begin{array}{ccc} -s\varphi \cdot c\vartheta & -s\vartheta & -c\varphi \cdot c\vartheta \\ -s\varphi \cdot s\vartheta & c\vartheta & -c\varphi \cdot s\vartheta \\ c\varphi & 0 & -s\varphi \end{array}\right|.$$

Here, cφ=cos(φ), sφ=sin(φ), cϑ=cos(ϑ), sϑ=sin(ϑ); φ is a geodetic latitude; ϑ is a geodetic longitude.

The block diagram of proposed system is shown in Figure 3.

### 2.5. Common Signal Processing Algorithms in GNSS Receiver

In Section 2.3, it is marked that the actual input for algorithm are correlator samples (8) that are generated in the GNSS receiver using reference signals $s_{ref,j,I}^{i}\left(t_{k,l}\right)$ and $s_{ref,j,Q}^{i}\left(t_{k,l}\right)$. Reference signals require estimations of delays ${\tilde{\tau }}_{A_{j},k}^{i}$ and Doppler frequency shifts ${\tilde{\omega }}_{D_{\Sigma },j;k}^{i}$, $i=\bar{1,n}$, $j=\bar{2,3}$ in incoming GNSS signals that can be obtained with non-coherent DLL and FLL.

#### 2.5.1. Frequency Lock Loop

To simplify the Doppler shift estimation algorithm, we intentionally use the same reference Doppler frequency for all three antennas. A stochastic model is used for the Doppler shift dynamic in the form
(30)$$\omega_{D_{\Sigma };k}^{i}=\omega_{D_{\Sigma };k-1}^{i}+T\nu_{k-1}^{i},\;\nu_{k}^{i}=\nu_{k-1}^{i}+\xi_{\nu ;k-1}^{i}.$$
where $\xi_{\nu ;k-1}^{i}$ is a zero-mean white Gaussian noise process with variance $D_{\xi_{\nu }}$.

By defining the state vector as $x_{\omega ,k}^{i}=\left|\begin{array}{cc} \omega_{D_{\Sigma };k}^{i} & \nu_{k}^{i} \end{array}\right|$ we can write down (30) in vector form
(31)$$x_{\omega ,k}^{i}=F_{\omega }x_{\omega ,k-1}^{i}+G_{\omega }\xi_{\nu ;k-1}^{i},\;\mathrm{where}\;F_{\omega }=\left|\begin{array}{cc} 1 & T\\ 0 & 1 \end{array}\right|,\;G_{\omega }=\left|\begin{array}{c} 0\\ 1 \end{array}\right|.$$

The filter for this process can be described by the following equation
(32)$${\hat{x}}_{\omega ,k}^{i}={\tilde{x}}_{\omega ,k}^{i}+Ku_{Discr,\omega ;k}^{i},\;{\tilde{x}}_{\omega ,k}^{i}=F_{\omega }{\hat{x}}_{\omega ,k-1}^{i},$$
where $u_{Discr,\omega ;k}$ is the output process of frequency discriminator [6]
(33)$$u_{Discr,\omega ;k}^{i}=\sum_{j=1}^{3}\left(I_{j,k}^{i}\left({\tilde{\omega }}_{D_{\Sigma },k}^{i}\right)\frac{\partial I_{j,k}^{i}\left({\tilde{\omega }}_{D_{\Sigma },k}^{i}\right)}{\partial \omega_{D_{\Sigma }}^{i}}+Q_{j,k}^{i}\left({\tilde{\omega }}_{D_{\Sigma },k}^{i}\right)\frac{\partial Q_{j,k}^{i}\left({\tilde{\omega }}_{D_{\Sigma },k}^{i}\right)}{\partial \omega_{D_{\Sigma }}^{i}}\right),$$
where $I_{j,k}^{i}\left({\tilde{\omega }}_{D_{\Sigma },k}^{i}\right)$, $Q_{j,k}^{i}\left({\tilde{\omega }}_{D_{\Sigma },k}^{i}\right)$ are defined by (8). Derivatives of these components are described in [6] as
(34)$$\frac{\partial I_{j,k}^{i}\left({\tilde{\omega }}_{D_{\Sigma },k}^{i}\right)}{\partial \omega_{D_{\Sigma }}^{i}}=-\sum_{l=1}^{M}y_{A_{j};k,l}h_{rc}^{i}\left(t_{k,l}-{\tilde{\tau }}_{A_{j},k}^{i}\right)\left(l-1\right)T_{d}\sin\left(\omega_{0}t_{k,l}+{\tilde{\omega }}_{D_{\Sigma },j;k}^{i}\left(l-1\right)T_{d}+\varphi_{rg;k,l}\right),\;\;\frac{\partial Q_{j,k}^{i}\left({\tilde{\omega }}_{D_{\Sigma },k}^{i}\right)}{\partial \omega_{D_{\Sigma }}^{i}}=\sum_{l=1}^{M}y_{A_{j};k,l}h_{rc}^{i}\left(t_{k,l}-{\tilde{\tau }}_{A_{j},k}^{i}\right)\left(l-1\right)T_{d}\cos\left(\omega_{0}t_{k,l}+{\tilde{\omega }}_{D_{\Sigma },j;k}^{i}\left(l-1\right)T_{d}+\varphi_{rg;k,l}\right).$$

The vector of loop filter coefficients K is calculated by known formulas [6] from the required bandwidth.

#### 2.5.2. Delay Lock Loop

This uses a stochastic model of delay dynamic in the form
(35)$$\tau_{k}^{i}=\tau_{k-1}^{i}-\frac{\omega_{D_{\Sigma };k-1}^{i}T}{2\pi f_{0}}+\xi_{\tau ;k-1}^{i},$$
where $\xi_{\tau ;k-1}^{i}$ is a zero-mean white Gaussian noise with the variance $D_{\xi_{\tau }}$, $f_{0}$ is a signal’s carrier frequency.

Here, we consider $\omega_{D_{\Sigma };k-1}^{i}$ as a partially determined process, that can be obtained from FLL output: $\omega_{D_{\Sigma };k-1}^{i}={\tilde{\omega }}_{D_{\Sigma };k-1}^{i}-\delta_{\omega ,k-1}$, where $\delta_{\omega ,k-1}$ is an FLL tracking error.

The filter for process (35) is described by Equations [6]
(36)$${\hat{\tau }}_{k}^{i}={\tilde{\tau }}_{k}^{i}+K_{\tau }u_{Discr,\tau ;k}^{i},\;{\tilde{\tau }}_{\omega ,k}^{i}={\hat{\tau }}_{k-1}^{i}-\frac{{\tilde{\omega }}_{D_{\Sigma };k-1}^{i}T}{2\pi f_{0}},$$
where ${\tilde{\omega }}_{D_{\Sigma };k-1}^{i}$ is the extrapolated Doppler shift estimation for the i-th NS produced by the tracking system (32),
(37)$$u_{Discr,\omega ;k}^{i}=\sum_{j=1}^{3}\frac{\partial X_{j,k}^{i}\left({\tilde{\tau }}_{k}^{i}\right)}{\partial \tau^{i}},$$
(38)$$X_{j,k}^{i}\left({\tilde{\tau }}_{k}^{i}\right)=\sqrt{{\left(I_{j,k}^{i}\left({\tilde{\tau }}_{k}^{i}\right)\right)}^{2}+{\left(Q_{j,k}^{i}\left({\tilde{\tau }}_{k}^{i}\right)\right)}^{2}}.$$

Equation (37) is written in supposition that the ranging code delay is approximately same for all three antennas. The derivatives in Equation (37) are computed as finite differences using early and late correlators [6,26]. The loop filter coefficient $K_{\tau }$ is calculated by a known formula [6] from the required bandwidth.

#### 2.5.3. Amplitude Estimator

Signal’s amplitude $A^{i}$ is required for the calculation of the discriminator’s slope $S_{discr}^{i}$ in (21). It can be shown with (9) that metric $X_{j,k}^{i}\left({\tilde{\tau }}_{k}^{i}\right)$ from (38) is equal to $\frac{A^{i}M}{2}$ with precision to noise term. Therefore, we can estimate amplitude with the following first-order filter
(39)$${\hat{A}}_{k}^{i}={\hat{A}}_{k-1}^{i}\cdot \left(1-\beta \right)+\beta \cdot \frac{2}{3M}\sum_{j=1}^{3}X_{j,k}^{i}\left({\tilde{\tau }}_{k}^{i}\right)$$
where β is the filter coefficient.

## 3. Simulation and Results

To evaluate the performance of the proposed system, the simulation model has been developed. The model includes user dynamics, GNSS GLONASS constellation (coordinates and velocities of satellites), 3x antenna model, three-axis MEMS gyro, correlator blocks and various tracking algorithms, as well as GNSS receiver navigation algorithms. All the processes in the model, i.e., correlator and gyro outputs, are simulated at a time step of *T* = 1 ms. The additional accumulation of I/Q components was performed with a variable time interval of an integer number of milliseconds.

### 3.1. Antenna Geometry

In the GNSS attitude determination system, the error depends on baseline length and the geometry of antenna. The antenna geometry chosen for simulation is an equal-sided triangle with a side length of 1 m. Hence, the attitude errors presented below are given for baseline length 1 m. Antenna geometry and body axes (*R,P,Y*) alignment are shown in Figure 4a.

### 3.2. GNSS Constellation

The attitude error also depends on the geometry of visible GNSS constellation. Only GLONASS constellation has been implemented. All signal characteristics correspond to L1OC signals. The sky view for the visible constellation is shown in Figure 4b. The simulation scenario included eight satellites with a geometric dilution of precision (GDOP) approximately equal to 2.5.

### 3.3. User Dynamics

Simulation scenario included the rotation of an antenna system around a vertical axis with an angular rate of about 360 °/s and sinusoidal tilts around the roll axis. The corresponding dynamics of roll, pitch and yaw angles are shown in Figure 5. The angular rates in RPY frame are shown in Figure 6. There was no transitional movement in the scenario.

### 3.4. Refernce Oscillator Dynamics

Commonly used TCXOs as the sources of time and frequency for GNSS receivers introduce severe dynamic disturbances for tracking loops. When it comes to the calculation of receiver’s sensitivity, only a proper account for the reference oscillator’s phase drift allows relevant results. Therefore, the recorded phase drift in real TCXO was used. The recording technique is described in [25]. The sample rate of phase drift process is 10 ksps. The TCXO is Morion’s GK-206TK. The experimental Allan deviation of this TCXO is shown in Figure 7a. The sample frequency drift process over one minute intervals is shown in Figure 7b. Frequency drift is transformed to the apparent Doppler’s velocity drift with $\left(c/f_{0}\right)$ coefficient, where c is the speed of light and $f_{0}$ is the nominal quartz frequency.

### 3.5. Gyroscope Parameters

A low-cost MEMS IMU MPU9250 from InvenSense is chosen as the prototype for gyro simulation. Its error parameters are shown in Table 1.

### 3.6. I/Q Adaptive Accumulation

The accumulation time for I/Q components in (8) determines the time interval for equivalent observations (19) in the integration filter. The lower this interval, the lower the prediction errors and higher the attitude accuracy. On the other hand, the less accumulation time for I/Q components in (8), the lower slope $S_{discr}^{i}$ and the higher effect of the noise $n_{j,k}^{i}$ at the discriminator output (18). Therefore, it is necessary to adjust the accumulation time ($MT_{d}$) for specific SNR to reach an optimum between noise and prediction errors. It is supposed that the accumulation time $T_{acc}$ is equal to the integer number of milliseconds, i.e., $T_{acc}=N_{acc}T$, where T = 1 ms, and $N_{acc}$ is computed by the following empirical formula
$$N_{acc}=\mathrm{round}\left(10^{0.1\left(50-C/N_{0}\right)}\right),$$
where $C/N_{0}$ is the estimate of average SNR in dBHz. $N_{acc}$ is limited in the range of 1…100.

### 3.7. Integration Filter Convergence and Attitude Precision

The attitude error process showing a convergence of the integration filters is presented in Figure 8. The initial attitude errors are about 2°. The front-end phase bias errors are $\delta_{A_{2}/A_{1}}$ = 12° and $\delta_{A_{3}/A_{1}}$ = 17°. The SNRs for received signals are within the typical range 35–40 dBHz depending on satellite elevation angle and antenna tilt. The accumulation time for I/Q components and discriminator processing interval is $T_{acc}$ = 22 ms. It is seen that the convergence time is about 20 s. The attitude errors are within ±7 arcmin.

The convergence process of relative front-end phase biases $\delta_{A_{2}/A_{1}}$ and $\delta_{A_{3}/A_{1}}$ is shown in Figure 9. Here, $\delta \psi_{2}\equiv \delta_{A_{2}/A_{1}}$ and $\delta \psi_{3}\equiv \delta_{A_{3}/A_{1}}$. One can see that these biases are estimated from initial errors 12° and 17° down to 0.5° precision.

### 3.8. Sensitivity and Attitude Precision in Range of SNR

To evaluate the attitude precision in a range of SNR and determine the lowest working SNR (sensitivity) for the proposed system, the following scenario has been applied. The system was operated for 20 s at SNR = 45–50 dBHz to reach a stationary state. Next, a slow degradation of SNR has been performed at the rate of 0.5 dBHz per second. The attitude error process for roll, pitch and yaw angles is shown in Figure 10. The SNR estimates for visible satellites are shown in Figure 11.

It can be seen that attitude error growth is non-linear. When SNR falls from 45–50 down to 18–23 dBHz, the attitude error increases adequately from ±3 to ±10 arcmin. Next, errors rapidly become ±23 arcmin and continue growing to ±1°. At this error level, signal tracking is totally lost. This happens at SNR 15–18 dBHz. Hence, the sensitivity of the proposed system is about 18 dBHz.

The rapid growth in attitude error at 55–57 s is connected with the loss of signal tracking. This can be seen from Figure 12, where the number of tracked signals during scenario is shown.

It should be noted that a loss of tracking happens at FLL/DLL, as described in Section 2.5. Therefore, the sensitivity of the whole system is limited by non-coherent tracking loops used for the tracking of the ranging code’s delay and Doppler frequency (including reference oscillator’s frequency drift). The deeply integrated phase-difference tracking system is apparently more sensitive.

The process of adaptation of accumulation time $T_{acc}$ is shown in Figure 13. There is actually no adaptation after the 34th second, when SNR is below 30 dBHz.

The Kalman filter’s error covariance well reflects actual attitude errors. This allows covariance estimate to be used as a measure of the resulting attitude’s precision. By using the Kalman filter’s error covariance, the average attitude RMS error $\sigma_{RPY}$ is obtained as a function of SNR ($C/N_{0}$). This dependency is shown in Figure 14. The average attitude RMS error is introduced as $\sigma_{RPY}=\sqrt{\left(\sigma_{R}^{2}+\sigma_{P}^{2}+\sigma_{Y}^{2}\right)/3}$, where $\left(\sigma_{R}^{2}+\sigma_{P}^{2}+\sigma_{Y}^{2}\right)\approx \left(E_{1,1}+E_{2,2}+E_{3,3}+E_{4,4}\right)/4$.

### 3.9. Attitude Error during GNSS Outage

When GNSS signals are not available (i.e., under the bridges or in tunnels) it is still possible to estimate attitude by using a gyroscope. This is done automatically in the integration filter. In this case attitude error will be growing due to gyroscope errors. Nevertheless, in the proposed design, gyro errors are partially estimated according to observability conditions and compensated for during GNSS outage, making the error growth as low as possible.

To evaluate the attitude growth rate during GNSS outage, a Monte-Carlo simulation of 20 tests is performed. In each test system, operated normally at SNR = 35–40 dBHz for 15 s, GNSS outage was simulated by setting the SNR to −100 dBHz. The outage duration is 60 s. Randomization of gyro errors and front-end noises for each test is applied. The resulting attitude errors are shown in Figure 15. It can be seen that yaw error grows to about ±2.5°, roll and pitch errors grow by about ±1° per minute. The average attitude RMS error grows with a rate of about 1° per minute. These errors can be considered as low for the chosen type of MEMS gyro.

## 4. Discussion

### 4.1. Sensitivity

By designing the proposed deeply integrated algorithm for a GNSS/gyro attitude determination system, we focused on improving sensitivity, i.e., providing attitude at the lowest possible SNR. As follows from simulation results, we have achieved an attitude determination error of about ±60 arcmin (1 m baseline) at the lowest possible SNR of 18 dBHz. Comparing this result with analogous result obtained from a standard algorithm, in which carrier phases from three antennas are measured by using separate PLLs, then phase differences are calculated and used for attitude determination without using a gyroscope [6,20]. In Figure 16, the analytical curve of angular root-mean-square error (RMSE) vs. SNR for standard algorithm with 1 m baselines is shown. Compared to Figure 14, one can see that an angular RMSE of 5 arcmin is reached at an SNR of about 22 dBHz in the proposed algorithm, while in the standard algorithm this point corresponds to SNR about 38 dBHz. Therefore, we have a sensitivity improvement of about 16 dB in the proposed algorithm.

The result is achieved due to deeply integrated gyroscope/phase-difference tracking system, in which the transitional user movement and referenced oscillator’s frequency drift are removed from phase difference dynamics. Most of the remained process dynamic is compensated for with gyro. The resulting sensitivity of the whole system is limited by the non-coherent tracking loops used for tracking the ranging code’s delay and the Doppler frequency of signals coming in the reference antenna. Hence, the further sensitivity improvement can be achieved by:Using all three antennas for the tracking of common Doppler frequency and code delay (+5 dB, approximately);Using deep integration of non-coherent tracking loops with INS and ultra-stable reference oscillator;Using pilot GNSS signals for tracking at low SNR.

### 4.2. Precision

The achieved (in simulation) attitude accuracy is about ±7–8 arcmin at SNR of 35–40 dBHz with 1 m baselines between antennas and using a low-cost MEMS gyro. These errors should not be related seriously because they are obtained without accounting for multipath phase errors that are the main source of error for the GNSS attitude. We also did not consider the wind-up effect and phase pattern of GNSS antennas, which are subject to specific hardware.

### 4.3. Ambiguity Resolution

Taking a glance at (21), one can see that the reference value of phase difference ${\tilde{\psi }}_{A_{j};k}^{i}$ used in discriminator (17) is not ambiguous—it is reconstructed directly from attitude and known baseline. The error signal at discriminator output is bounded within ±1 radian because it is proportional to $\sin\left(\Delta_{\psi }\right)$, where $\Delta_{\psi }=\psi_{A_{j};k}^{i}-{\tilde{\psi }}_{A_{j};k}^{i}$ is the true tracking error. Therefore, the equivalent linear observations (19) (that can be derived from (18) as ${\tilde{y}}_{\psi_{A_{j}};k}^{i}=u_{Discr,\psi_{A_{j}};k}^{i}+S_{discr}^{i}\left({\tilde{\psi }}_{A_{j};k}^{i}+{\tilde{\delta }}_{A_{j}/A_{1}}\right)$) are not ambiguous. Hence, the ambiguity resolution procedure is required only once—for initial attitude determination and, occasionally, after long GNSS outages with more than a 1–2 min duration, while in most analogous systems, this computationally heavy procedure must be performed at each step. For initial ambiguity resolution, well-known Lambda, BC-Lambda or C-Lambda methods can be used, as well as initial alignment from accelerometers and magnetic compass.

### 4.4. Cycle Slips

When tracking a GNSS signal with common PLL, the cycle slip may occur because of signal interruption (due to multipath or shadowing) or because of a rapid phase jerk due to user dynamics or reference quartz oscillator stress. Cycle slip means missing a number of integer phase cycles from the observed carrier phase. This is the common problem of all GNSS RTK and attitude determination systems. In the proposed system, phase difference is analogue to carrier phase. As opposed to the general GNSS receiver design, there are no separate loops tracking phase differences in this system, therefore no cycle slip in such a loop can occur. In other words, we do not track the carrier phase, so there is no place for a cycle slip to occur. The signal degradation effects mentioned above influence the system in different ways. Due to vector tracking, the signal degradation in only one channel may lead (in worst case) to a bias of discriminator (17) output. This will cause a minor increase in attitude error, but the tracking will be stabilized due to the normal work of other channels. This makes the proposed system less vulnerable to the effects that cause cycle slips in a normal GNSS receiver.

## 5. Conclusions

This article described the development of the deeply integrated algorithm of attitude determination by using GNSS signals from three spaced antennas and a three-axis gyro. The optimal nonlinear filtering theory was used. Deep integration technology (methodology) provided a sensitivity (and hence jammer immunity) improvement. In the synthesized integration algorithm, the direct estimate of inter-antenna phase difference is used in each pair of antennas. For the transitional movement tracking, non-coherent FLL and DLL are used. The block diagram of the synthesized deeply integrated GNSS/gyro attitude determination system is provided.

The dependency of attitude estimation error on SNR is analyzed using a simulation model. It is shown that, even with a low-cost MEMS gyro, the attitude RMS error of 5 arcmin is achieved at SNR = 22–23 dBHz, 16 dB lower than that of the standard algorithm with the same baseline of 1 m. The lowest SNR at which attitude determination is maintained for the proposed system is 18 dBHz, the attitude RMS error in this case is about 20 arcmin. This SNR is equivalent to a situation when the noise jammer is present with a jammer-to-signal ratio J/S = 44 dB within a 2 MHz system bandwidth. This implies that the proposed system improves jammer immunity by 10–14 dB in comparison to a standard coherent GNSS receiver with attitude function [6,24].

## Figures and Tables

**Figure 1 sensors-20-02203-f001:**
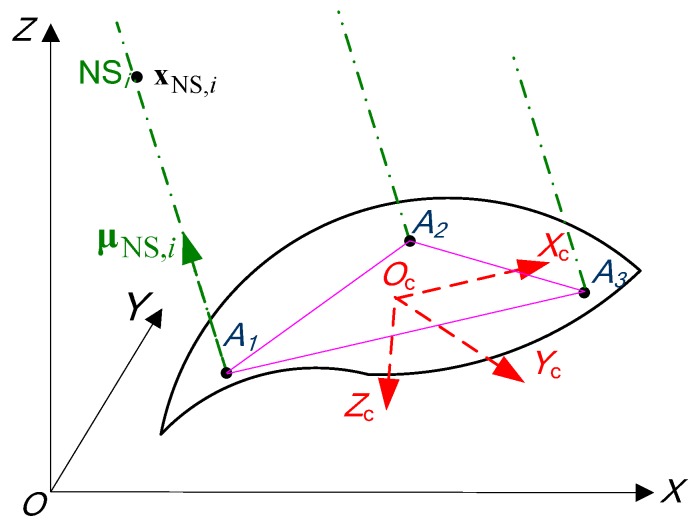
Geometry of baselines and reference frames.

**Figure 2 sensors-20-02203-f002:**
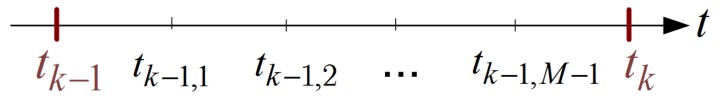
Double-indexed time scale.

**Figure 3 sensors-20-02203-f003:**
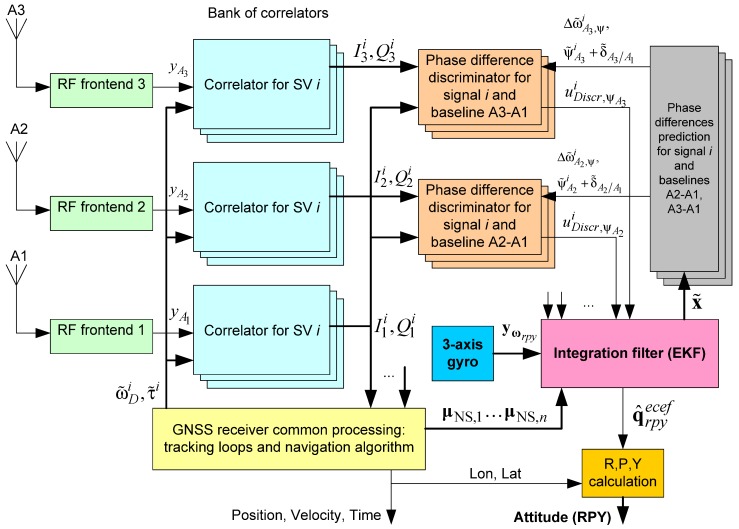
Block diagram of deeply integrated GNSS/Gyro attitude determination system.

**Figure 4 sensors-20-02203-f004:**
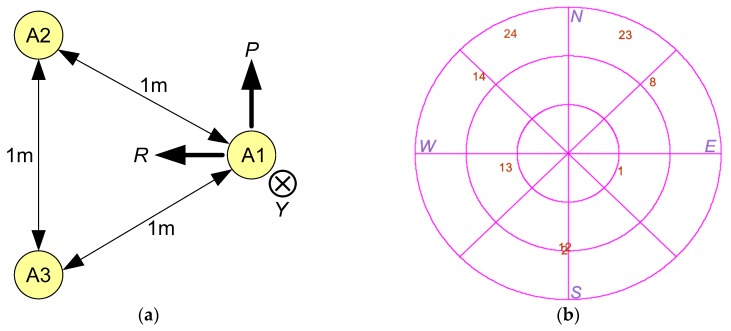
Geometry issues: (**a**) Antenna geometry and body frame axes (*R,P,Y*); (**b**) Sky view for GNSS’ visible constellation.

**Figure 5 sensors-20-02203-f005:**
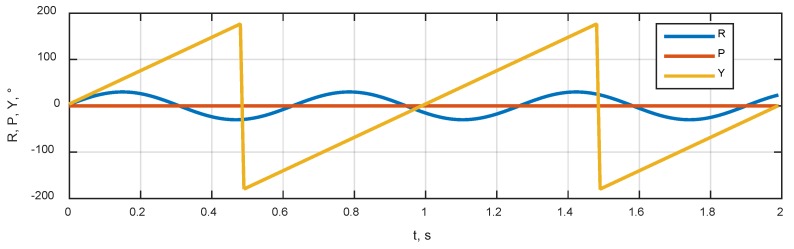
The dynamics of Roll, Pitch and Yaw angles.

**Figure 6 sensors-20-02203-f006:**
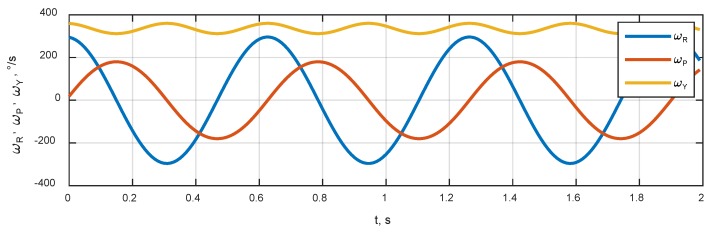
Angular rates in RPY frame.

**Figure 7 sensors-20-02203-f007:**
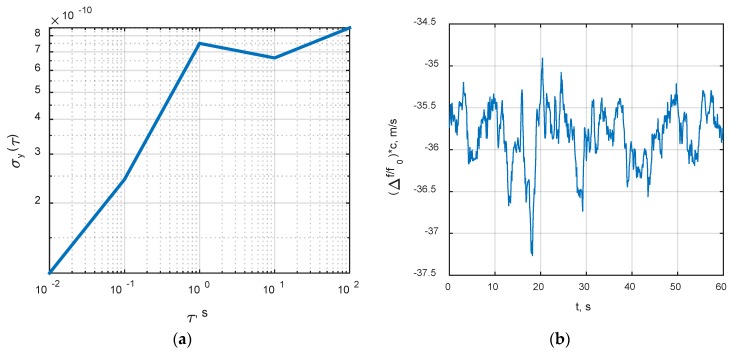
The characteristics of TCXO GK-206TK used in simulation: (**a**) Measured Allan deviation; (**b**) Frequency drift process in units of apparent Doppler’s velocity.

**Figure 8 sensors-20-02203-f008:**
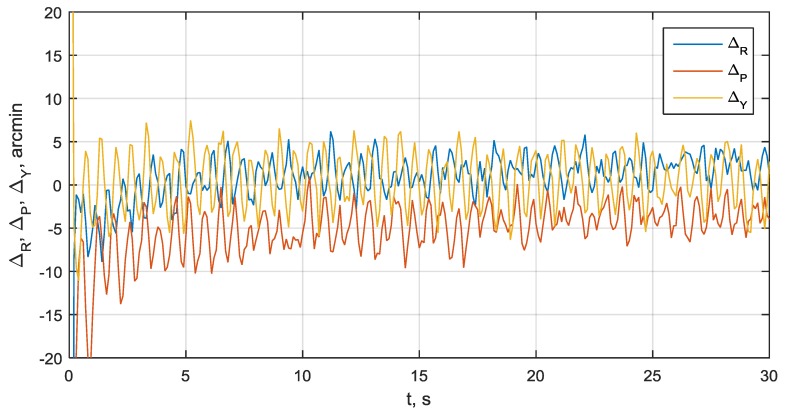
Roll, pitch and yaw errors convergence process at SNR 35–40 dBHz.

**Figure 9 sensors-20-02203-f009:**
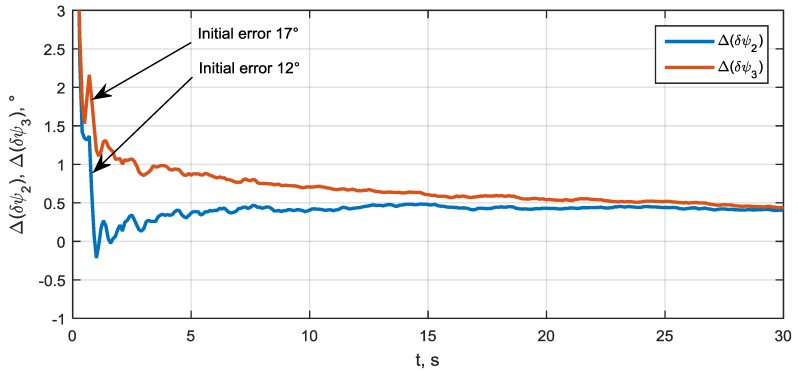
Front-end phase bias estimation errors.

**Figure 10 sensors-20-02203-f010:**
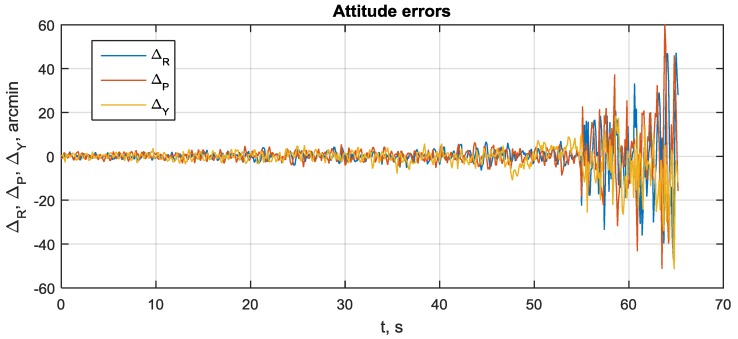
Roll, pitch and yaw errors during SNR degradation from 50 to 20 dBHz.

**Figure 11 sensors-20-02203-f011:**
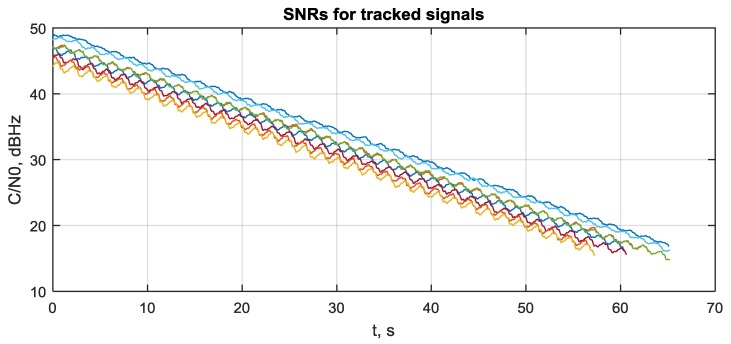
Simulated SNR degradation.

**Figure 12 sensors-20-02203-f012:**
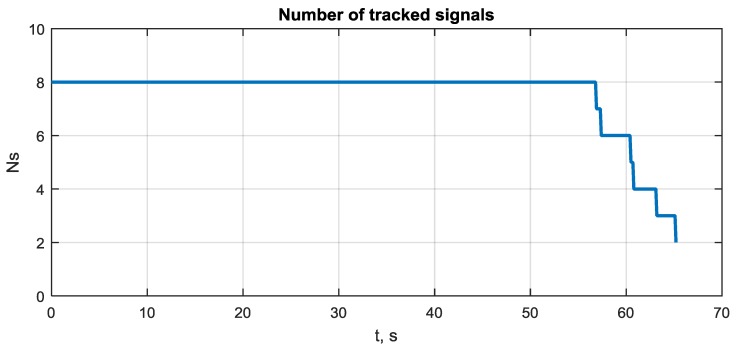
Number of tracked signals during simulation scenario.

**Figure 13 sensors-20-02203-f013:**
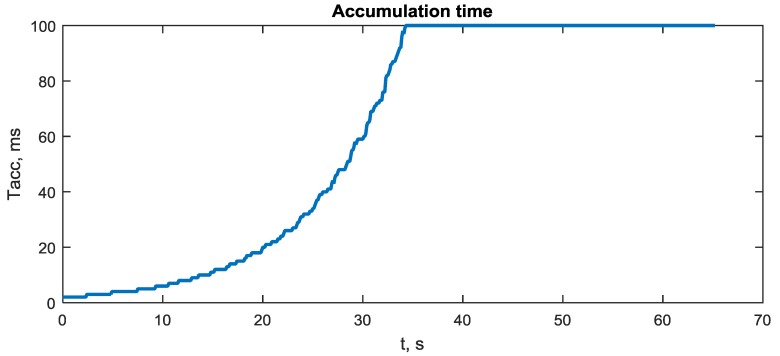
Accumulation time adaptation process induced by the degradation of SNR.

**Figure 14 sensors-20-02203-f014:**
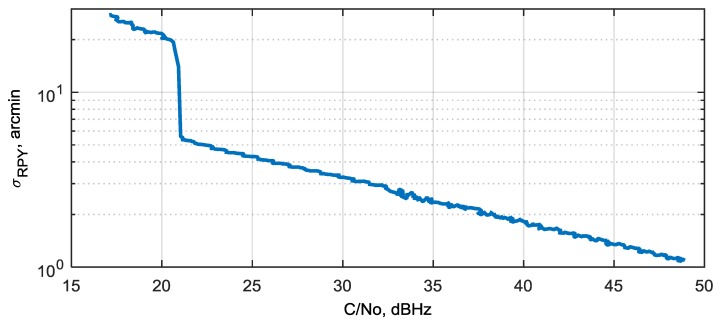
Average attitude RMS error vs. SNR.

**Figure 15 sensors-20-02203-f015:**
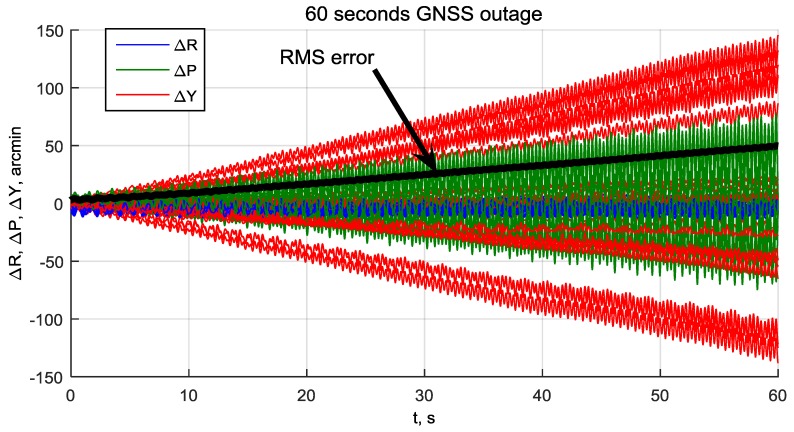
Roll, pitch and yaw errors during GNSS outage.

**Figure 16 sensors-20-02203-f016:**
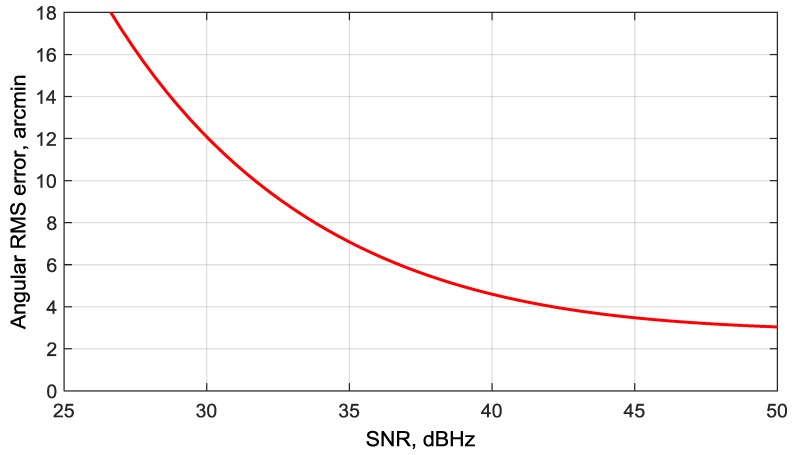
Angular RMS error vs. SNR for standard attitude determination algorithm.

**Table 1 sensors-20-02203-t001:** Gyroscope errors.

Parameter	Value
From-run-to-run bias stability	±2 (°/s)
Angular random walk (standard deviation)	7 × 10^−3^ (${}^{\circ}/\sqrt{s}$)
Flicker noise (standard deviation)	6.6 × 10^−3^ (°/s)
Scale factor error	±10^−2^
Axis-to-axis misalignment	±0.05°
Linear acceleration effect on bias	±0.05 (°/s/g)

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
