# Peer review of "Deeply Integrated GNSS/Gyro Attitude Determination System"

_sensors, 2020, doi:10.3390/s20082203_

Round 1

Reviewer 1 Report

  1. Generally, in GNSS attitude determination system, the error depends on Baseline length and geometry of satellites and antenna (including the mentioned factor in the paper). [Chansik Park, Dong-Hwan Hwang and Sang-Jeong Lee, “Error Analysis of 3 Dimensional GPS Attitude Determination System,” International Journal of Control and Systems, Vol. 4, No. 4, pp.480-485, Aug., 2006]
  2. This paper mentioned ambiguity resolution in introduction, but I can not find the requirement of ambiguity resolution in the proposed method. In Discussion, it briefly mentioned, but still hard to understand. [For attitude determination, instead of LAMBDA, BC-Lambda or C-Lambda are more powerful methods)
  3. GNSS signal attenuation (low SNR) may cause the quality degradation of Attitude determination system, However, cycle slip detection and repair are also vulnerable points to be considered.
  4. Third eq. in (13) has typo-error., In line 210, in (11) may be read in (14).
  5. Comparison with other methods are required to insist the strength of the proposed method.
  6. The conclusion is too short.

Author Response

Point 1: Generally, in GNSS attitude determination system, the error depends on Baseline length and geometry of satellites and antenna (including the mentioned factor in the paper). [Chansik Park, Dong-Hwan Hwang and Sang-Jeong Lee, “Error Analysis of 3 Dimensional GPS Attitude Determination System,” International Journal of Control and Systems, Vol. 4, No. 4, pp.480-485, Aug., 2006]

Response 1:  All the estimates are given for baseline 1 m. This fact is stressed in new edition in section 3.1, discussion and conclusion.

Point 2: This paper mentioned ambiguity resolution in introduction, but I can not find the requirement of ambiguity resolution in the proposed method. In Discussion, it briefly mentioned, but still hard to understand. [For attitude determination, instead of LAMBDA, BC-Lambda or C-Lambda are more powerful methods)

Response 2: In new edition this question is more thoroughly considered in introduction and discussion sections.

Point 3: GNSS signal attenuation (low SNR) may cause the quality degradation of Attitude determination system, However, cycle slip detection and repair are also vulnerable points to be considered.

Response 3: In new edition this question is considered in discussion section.

Point 4: Third eq. in (13) has typo-error., In line 210, in (11) may be read in (14).

Response 4: Corrected in new edition.

Point 5: Comparison with other methods are required to insist the strength of the proposed method.

Response 5: The comparison with common method is added to discussion, section 4.1.

Point 6: The conclusion is too short.

Response 6: The conclusion is enhanced with review what is done and what is achieved.

Reviewer 2 Report

  1. Check the spelling and English throughout the paper, like "trough".
  2. The introduction is not strong. Discuss more on the innovations of your method and the scenarios that your method could be implemented.
  3. Suggest to put the overall framework of the algorithm at the front of Section 2.
  4. There is nothing novel in Section 2.1 - 2.3. Suggest to cite the references and just write down the important formulas, to make it more concise.
  5. Some comparisons of the results with and without gyros are expected.
  6. Rewrite the conclusion part.

Author Response

Point 1: Check the spelling and English throughout the paper, like "trough".

Response 1: Spelling and grammar checks are performed in new edition.

Point 2: The introduction is not strong. Discuss more on the innovations of your method and the scenarios that your method could be implemented.

Response 2: The words about that are added to introduction.

Point 3: Suggest to put the overall framework of the algorithm at the front of Section 2.

Response 3: Suggestion is kindly accepted, the framework is added before subsection 2.1

Point 4: There is nothing novel in Section 2.1 - 2.3. Suggest to cite the references and just write down the important formulas, to make it more concise.

Response 4: Sections 2.1 - 2.3 are shortened as suggested

Point 5: Some comparisons of the results with and without gyros are expected.

Response 5: The comparison is added to discussion, subsection 4.1

Point 6: Rewrite the conclusion part.

Response 6: The conclusion is totally rewritten and enhanced in new edition

Reviewer 3 Report

The structure of the article is considered and clear. In the introduction, the background and comprehensive review of the problem's literature were presented. The Authors presents reference frames for the three antennas and the gyro system, which is the main problem of the article. Signals and processing algorithms are described in mathematical view. Assumptions for simulations are closely described. Results of research have been presented in graphic form. Conclusions, on the basis of the research, are clear.

Line 192: in first equation uthors use fraction bar, in next one oblique (/)
Line 344: in table 1 "x" should rather be replaced by point: not 7x103, rather 7·103 - "x" is used in vector multiplication
Lines 256, 303. 448, 449, 459: there are chinese letter in equations - probably printing error durin creating pdf file.

Author Response

Thank you very much for provided review!

Regarding your remarks:

Line 192: in first equation authors use fraction bar, in next one oblique (/)

Changed to fraction bar in both equations

Line 344: in table 1 "x" should rather be replaced by point: not 7x103, rather 7·103 - "x" is used in vector multiplication

Yes, "x" is replaced by multiplication point.

Lines 256, 303. 448, 449, 459: there are chinese letter in equations - probably printing error durin creating pdf file.

I could not reproduce this error. All nearby computers including my cell phone show normal equations.

The conclusions part will be detailed and enhanced.